# *switch*-GLAT: Multilingual Parallel Machine Translation via Code-Switch Decoder

**Zhenqiao Song**[1], **Hao Zhou**[*1], **Lihua Qian**[1], **Jingjing Xu**[1], **Shanbo Cheng**[1], **Mingxuan Wang**[1], **Lei Li**[2]

[1]ByteDance AI Lab, Shanghai, China    [2]University of California, Santa Barbara

`{songzhenqiao,zhouhao.nlp,qianlihua}@bytedance.com, lilei@ucsb.edu`
`{chengshanbo,wangmingxuan.89}@bytedance.com, jingjingxu@pku.edu.cn`

## Abstract

Multilingual machine translation aims to develop a single model for multiple language directions. However, existing multilingual models based on Transformer are limited in terms of both translation performance and inference speed. In this paper, we propose *switch*-GLAT, a non-autoregressive multilingual machine translation model with a code-switch decoder. It can generate contextual code-switched translations for a given source sentence, and perform code-switch back-translation, greatly boosting multilingual translation performance. In addition, its inference is highly efficient thanks to its parallel decoder. Experiments show that our proposed *switch*-GLAT outperform the multilingual Transformer with as much as $0.74$ BLEU improvement and $6.2x$ faster decoding speed in inference.

## 1 Introduction

Neural machine translation (NMT) is the current state-of-the-art approach (Bahdanau et al., 2014; Sutskever et al., 2014) for machine translation in both academia (Bojar et al., 2017) and industry (Hassan et al., 2018). Recent works (Firat et al., 2016; Johnson et al., 2017; Aharoni et al., 2019; Lin et al., 2020) extend the approach to support multilingual translation, i.e. training a single model that can translate across multiple language directions. Multilingual models are appealing for several reasons. First, they can reduce the online translation service number, enabling simpler deployment (Arivazhagan et al., 2019) when plenty of translation directions are required. Additionally, multilingual training makes it possible to transfer knowledge from high-resource languages to low-resource ones, thus improving the translation quality of low-resource directions (Zoph et al., 2016; Johnson et al., 2017; Wang & Neubig, 2019).

However, most multilingual NMT systems are built upon the autoregressive architecture, which translates from left to right and thus is not efficient enough in terms of translation speed. Such efficiency problem is more serious in multilingual setting because all translation directions suffer from this slow inference speed.

A straightforward solution to improve the multilingual translation efficiency is to develop multilingual non-autoregressive translation (NAT). NAT generates translation outputs in parallel (Gu et al., 2018), which leads to significantly faster translation speed. Thanks to the recent progress of NAT (Ghazvininejad et al., 2019; Gu et al., 2019; Deng & Rush, 2020), current state-of-the-art NAT models have achieved comparable BLEU scores (Li et al., 2018; Wei et al., 2019; Qian et al., 2020) with their auto-regressive counterparts. Among them, the glancing transformer (GLAT) proposed by Qian et al. (2020) is a representative work, which even outperforms many strong autoregressive translation systems in BLEU score on German-English translation task of WMT21 (Qian et al., 2021).

In this paper, we argue that multilingual NAT models is not only superior in efficiency, but also can achieve better multilingual translation accuracy, due to its capability of generating high quality code-switched translations (Lin et al., 2020; Yang et al., 2020; Jose et al., 2020). In particular, we propose *switch*-GLAT, a carefully designed multilingual version of GLAT, which can outperform multilingual Transformer in both speed and translation quality. Generally, the main idea of *switch*-GLAT is to

---

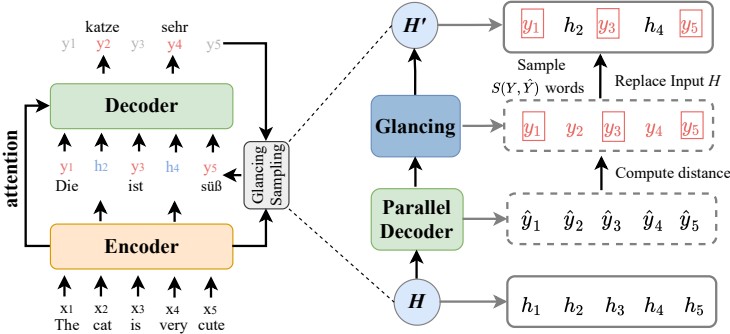

Figure 1: The overall architecture of GLAT. The left half shows the training process of GLAT while the right half details the glancing sampling strategy.

employ a *code-switch decoder*, which can generate *contextual code-switched* translations (instead of using dictionary for replacement) for a given source sentence, and then perform *code-switch back-translation* to boost the multilingual translation performance. In more details, the glancing sampling module in GLAT enables *switch*-GLAT to generate partial translation based on others. Then combined with its multilingual and non-autoregressive characteristics, *switch*-GLAT could output code-switched translations with the employment of *token-level language tags* instead of sentence-level ones in auto-regressive models. Ultimately, *switch*-GLAT can generate contextual translated words at arbitrary positions of the target sentence in arbitrary languages. This greatly improves the multilingual translation performance when we reverse the pairs of source to contextual code-switched target sentences, for training in a back-translation fashion (so called code-switch back-translation).

We conduct extensive experiments on 3 merged translation datasets: WMT with four language pairs (both close languages and distant ones) and WMT with 10 language pairs. *switch*-GLAT shows consistent improvements over autoregressive multilingual baselines on all datasets, validating that *switch*-GLAT can achieve better multilingual translation performance simultaneously with a faster decoding speed. We further evaluated the cross-lingual representations through word induction and sentence retrieval tasks. The results demonstrated the proposed code-switch back-translation benefits better-aligned cross-lingual representations.

## 2 BACKGROUND

**Multilingual Neural Machine Translation (MNMT)** Given a source sentence $X = \{x_1, x_2, ..., x_M\}$ with length $M$ and its target sentence $Y = \{y_1, y_2, ..., y_N\}$ with length $N$, MNMT leverages the standard bilingual neural machine translation models and extends the source and target inputs respectively with a source and target language token src and tgt. This results in $X' = \{\text{src}, x_1, x_2, ..., x_M\}$ and $Y' = \{\text{tgt}, y_1, y_2, ..., y_N\}$.

MNMT is generally modeled from $X'$ to $Y'$ with Transformer (Vaswani et al., 2017). Transformer consists of stacked encoder and decoder layers, which are jointly trained to maximize the conditional probability of $Y'$ given $X'$:

$$P(Y'|X') = \sum_{i=1}^{N} \log P(y_i|y_{<i}, X', \text{tgt}; \theta) \tag{1}$$

where $\theta$ are the trainable model parameters.

**Glancing Transformer** prposed by Qian et al. (2020) is a NAT architecture which achieves top results in machine translation with $8x \sim 15x$ speedup. It performs two-pass decoding in training but is still fully non-autoregressive in inference. In the first decoding pass, given the encoder $F_e$ and decoder $F_d$, $H_d^0 = \{h_1^0, h_2^0, ..., h_N^0\}$ is the decoder input either gathered from the encoder output using *soft copy* (Wei et al., 2019) or *full mask* (Ghazvininejad et al., 2019), and then $Y$ is predicted as:

$$\hat{Y} = F_d(H_d^0, F_e(X; \theta); \theta) \tag{2}$$

where $\theta$ are the trainable model parameters. Then glancing transformer (GLAT) adopts glancing sampling strategy to sample a subset of $Y$ according to its distance with $\hat{Y}$, thus resulting in $GS(Y, \hat{Y})$.

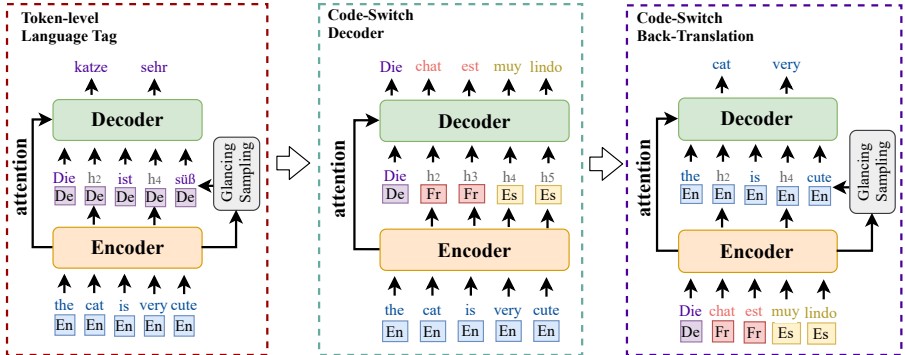

Figure 2: The overall architecture of *switch*-GLAT. The left module shows that GLAT is extended to a multilingual version with the token-level language tag, the middle one describes how the code-switched sentence is produced through the code-switch decoder, and the right module introduces the code-switch back-translation process, in which the generated code-switched target and its paired source is swapped to enhance training.

Finally, GLAT predicts the target sequence $Y$ based on this subset and source sentence $X$ in the second decoding pass as follows:

$$L_{\text{GLAT}} = - \sum_{y_t \in \mathbb{GS}(Y, \hat{Y})} \log P(y_t | \mathbb{GS}(Y, \hat{Y}), X; \theta) \tag{3}$$

where $H_d^0$ is updated accordingly and $\overline{\mathbb{GS}(Y, \hat{Y})}$ denotes the remaining subset of $Y$ after removing the sampled tokens. The overall architecture of GLAT is shown in Figure 1.

The twice decoding in training makes GLAT capable of predicting some partial translations based on others. Then combined with multilingual and non-autoregressive characteristics, GLAT has the potential to generate contextual code-switched outputs. In the next section, we will show how to modify the GLAT architecture to achieve this.

## 3 PROPOSED METHOD: *switch*-GLAT

This section will detail the proposed *switch*-GLAT. The training objective can be factorized into two parts. One aims to make the model have multilingual translation ability, and the other to achieve better-aligned cross-lingual representations to boost translation performance, respectively denoted as $L_{\text{multi}}$ and $L_{\text{csbt}}$. The overall objective can be formulated as follows:

$$L = L_{\text{multi}} + \lambda \cdot L_{\text{csbt}} \tag{4}$$

where $\lambda$ plays the role of a "temperature" to schedule the importance of code-switch back-translation. As the training progresses, $\lambda$ is gradually increased, which allows for more complex data instances to be involved, i.e. code-switched translations incorporating more languages and higher switching ratios. This process encourages model to align similar words from different languages into the same vector space, thus boosting machine translation performance. The overall architecture is shown in Figure 2.

### 3.1 CODE-SWITCH DECODER

*switch*-GLAT leverages the general parallel decoder of GLAT and extends it into a code-switch decoder by employing the token-level language tag. The code-switch decoder first gains multilingual translation ability through multilingual training, and then it can generate code-switched translations in arbitrary languages with the help of token-level language tag.

Specifically, given a multilingual corpora $\mathbb{D} = \{\mathbb{D}^l\}_{l=1}^L$ consisting of $L$ language pairs, the loss $L_{\text{multi}}$ is then defined as:

$$L_{\text{multi}} = \sum_{\mathbb{D}^l \in \mathbb{D}} \sum_{(X_j^l, Y_j^l) \in \mathbb{D}^l} \{L_{\text{tag}}(Y_j^l | X_j^l; \theta_M) + L_{\text{len}}^l(j)\}$$

$$L_{\text{len}}^l(j) = -P(L_j^l) \log \hat{P}(L_j^l | [F_e(X_j^l; \theta_M); E_{\text{src}}; E_{\text{tgt}}]; \theta_M) \tag{5}$$

where $\mathbb{D}^l = \{(X_j^l, Y_j^l)\}_{j=1}^{N_l}$ is a parallel corpus with size $N_l$ and $L_{\text{len}}^l(j)$ is the length prediction loss of the $j$-th pair. $P(L_j^l)$ is the real length distribution of target sentence, and $\hat{P}(L_j^l)$ is the predicted one based on the concatenation of encoder output as well as source and target language embeddings. $\theta_M$ are trainable model parameters. $L_{tag}$ is the accordingly updated GLAT training loss (Equation 3) incorporating the source and target language tags:

$$L_{\text{tag}}(Y_j^l|X_j^l; \theta_M) = - \sum_{y_t \in \mathbb{GS}(Y_j^l, \hat{Y}_j^l)} \log P(y_t|\mathbb{GS}(Y_j^l, \hat{Y}_j^l), X_j^l, \text{src}, \text{tgt}; \theta_M)$$

$$\hat{Y}_j^l = F_d(\tilde{H}_d^0, F_e(X_j^l, \text{src}; \theta), \text{tgt}; \theta_M)$$

(6)

To involve the indicative language tag, we add it to the first layer input and final layer output at each position of both encoder and decoder as follows:

$$\tilde{f}_i^0 = f_i^0 + E_{\text{src}}; \tilde{f}_i^K = f_i^K + E_{\text{src}}$$
$$\tilde{h}_j^0 = h_j^0 + E_{\text{tgt}}; \tilde{h}_j^K = h_j^K + E_{\text{tgt}}$$

(7)

where $f_i^0$ denotes the first encoder layer input at position $i$ and $h_j^0$ denotes the first decoder layer input at position $j$. Correspondingly, $f_i^K$ and $h_j^K$ denote the last layer output. src and tgt are respectively the source and target individual tags, while $E_{\text{src}}$ and $E_{\text{tgt}}$ are their corresponding representations. The overall prediction process of pair $(X_j^l, Y_j^l)$ is illustrated in the left module of Figure 2.

Through the multilingual training process, *switch*-GLAT can translate between different languages using the indicative language tags, of which the decoder is called code-switch decoder. It has the ability to generate contextual translated words in arbitrary languages due to its token-level characteristics.

## 3.2 CODE-SWITCH BACK-TRANSLATION

Thanks to the code-switch decoder, we can perform code-switch back-translation (CSBT), which is critical because it encourages model to align the produced words and the original ones into the same vector space according to their similar context information. Better-aligned cross-lingual representations benefit better translation performance.

Specifically, a subset $\mathbb{D}_S = \{(X_i, Y_i, l_i^{\text{src}}, l_i^{\text{tgt}})\}_{i=1}^{S}$ of size $S$ is first sampled, where $l_i^{\text{src}}$ and $l_i^{\text{tgt}}$ are respectively source and target languages of $i$-th pair. Then, $Y_i$ is masked with a given rate $P_M$, leading to $\tilde{Y}_i$. Subsequently, the masked positions of $\tilde{Y}_i$ can be decoded into a third randomly sampled language by leveraging the token-level language tag. Thus, the final decoded sequence $\hat{Y}_i$ consists of contextual tokens from mixed languages, which will in turn be taken as the source side input and the original source sentence as the target side input. This process results in a code-switch back-translation corpus $\mathbb{D}_C = \{(\hat{Y}_i, X_i)\}_{i=1}^{S}$, which is illustrated in the middle module of Figure 2.

The dynamically generated code-switch translations can augment data instance distribution to enhance model training as illustrated in the right module of Figure 2. Loss $L_{\text{csbt}}$ is defined as:

$$L_{\text{csbt}} = \sum_{(\hat{Y}_i, X_i) \in \mathbb{D}_C} L_{\text{tag}}(X_i|\hat{Y}_i; \theta_M)$$

(8)

As the training continues, the masked rate $P_M$ and number of mixed languages are gradually increased. Specifically, the value of $P_M$ is iterated from $0.1$ to $0.5$ with step size $0.1$ every 10 epochs. In the first iteration of $P_M$, the number of mixed languages is set to 1. Afterwards, it will be increased to one-third of the total. Through this process, abundant code-switched sentences can be generated, which helps to learn better-aligned cross-lingual representations.

## 3.3 SCHEDULED CODE-SWITCH TRAINING

The influence of multilingual training and code-switch back-translation is balanced by $\lambda$. As the training progresses, $\lambda$ is increased and more complicated data instances are involved, which means that code-switch back-translation becomes more important. We use the step function to evolve the value of $\lambda$:

$$\lambda(t) = f(t) + \lambda(0), \quad f(t) = \begin{cases} 0, & t < E, \\ 1, & t \geq E \end{cases}$$

(9)

where $\lambda(t)$ is the importance value at step $t$ and $E$ is a pre-defined changing point to incorporate code-switched translations. $\lambda(0)$ is the starting value and we set it to 0. As the training goes on, the code-switch back-translation and model training are iterated, leading to well-aligned cross-lingual representations and improved machine translation performance.

## 4 EXPERIMENTS

### 4.1 DATASETS

To better measure the effects of multilingualism at different levels, we test our proposed *switch*-GLAT on the following three merged datasets: (1) WMT-EDF: We collect 4 language pairs from WMT-14 English (En) ↔ German (De) and English (En) ↔ French (Fr). All three languages belong to Indo-European language family and are relatively *close* on linguistics. (2) WMT-EFZ: We also collect 4 language pairs from WMT-14 English (En) ↔ French (Fr) and WMT-17 English (En) ↔ Chinese (Zh), which are *distant* languages on linguistics and their relationships are more difficult to learn. (3) WMT-many: We also gather 10 language pairs from WMT-14 English (En) ↔ German (De), English (En) ↔ French (Fr), WMT-16 English (En) ↔ Russian (Ru), English (En) ↔ Romanian (Ro) and WMT-17 English (En) ↔ Chinese (Zh) to test *switch*-GLAT on more *diverse* language pairs.

### 4.2 BASELINE MODELS

We compare *switch*-GLAT against the following representative **multilingual baselines**: (1) **M-Transformer**: We set a multilingual transformer baseline, which also uses token-level language tags on both source and target sides. (2) **CLSR**: Zhang et al. (2021) present CLSR that learns to route between language-specific or shared pathways from the data itself to obtain the sharing structure. (3) **Adapter**: Bapna et al. (2019) propose to inject a lightweight adapter layer for each language in MNMT to extract some language-specific features. We reimplement Adapter since its code is not available currently. (4) **MNAT**: We also set a multilingual NAT baseline (Gu et al., 2018) modeled by adding token-level language tags to the vanilla NAT. All model settings are the same as *switch*-GLAT. Besides, we also report the performance of two typical **bilingual models**: (1) **Transformer**: We use the code released by Vaswani et al. (2017) to implement Transformer. (2) **GLAT**: GLAT is the backbone of our *switch*-GLAT and it achieves comparable results to Transformer. To better analyze the influence of different components in our *switch*-GLAT, we also conduct **ablation tests** as follows: (1) ***switch*-GLAT-w/o-glancing**: *switch*-GLAT removes glancing sampling strategy in training. (2) ***switch*-GLAT-w/o-CSBT**: *switch*-GLAT is trained without code-switch back-translation, which means the model is only trained on the golden parallel corpora. (3) ***switch*-GLAT-w/o-CCS**: *switch*-GLAT constructs the code-switched data by simply replacing aligned words using a bilingual dictionary instead of contextualized code-switching.

| Models | WMT-EDF | | | | | | WMT-EFZ | | | | |
|---|---|---|---|---|---|---|---|---|---|---|---|
| | En-De | De-En | En-Fr | Fr-En | Avg | Speed | En-Fr | Fr-En | En-Zh | Zh-En | Avg |
| *Bilingual models* | | | | | | | | | | | |
| Transformer | 27.77 | 31.55 | 38.80 | 37.35 | **33.86** | 1.3× | 38.80 | 37.35 | 23.60 | 24.05 | **30.95** |
| GLAT | 26.09 | 30.53 | 38.62 | 34.44 | 32.42 | 6.3× | 38.62 | 34.44 | 21.05 | 22.89 | 29.25 |
| *Multilingual models* | | | | | | | | | | | |
| M-Transformer | 25.84 | 31.57 | 38.52 | 36.03 | 32.99 | 1.0× | 38.06 | 35.07 | 20.76 | 22.19 | 29.02 |
| CLSR | 23.51 | 31.29 | 38.58 | 34.67 | 32.01 | 0.9× | 37.39 | 35.62 | 20.23 | 21.13 | 28.59 |
| Adapter | 22.19 | 29.56 | 40.72 | 35.88 | 32.08 | 0.9× | 40.13 | 35.02 | 19.87 | 21.29 | 29.08 |
| MNAT | 13.82 | 21.89 | 24.31 | 25.28 | 21.32 | 5.9× | 19.46 | 20.18 | 7.89 | 7.35 | 13.72 |
| *switch*-GLAT | 25.27 | 31.29 | 40.81 | 36.00 | **33.34** | 6.2× | 40.54 | 36.48 | 19.47 | 22.55 | **29.76** |
| – w/o glancing | 17.76 | 23.28 | 25.91 | 29.15 | 24.02 | 6.0× | 21.28 | 21.97 | 8.62 | 8.02 | 14.97 |
| – w/o CSBT | 24.29 | 29.03 | 36.45 | 34.09 | 30.97 | 6.2× | 35.17 | 33.65 | 18.32 | 20.37 | 26.87 |
| – w/o CCS | 24.72 | 29.51 | 37.32 | 34.17 | 31.43 | 6.1× | 35.96 | 33.61 | 18.45 | 20.83 | 27.21 |

Table 1: Translation performance (BLEU) on WMT-EDF/EFZ[1]. Avg means the average BLEU score.

---

[1]We updated all results in camera ready version using *multi-bleu.perl* script of Moses for evaluation because we misused *-lc* command in the original submission. Though the average BLEU scores were $0.0 \sim 0.8$ points lower than before, the overall conclusions are not changed.

| Models | WMT-many | | | | | | | | | | |
|---|---|---|---|---|---|---|---|---|---|---|---|
| | En-De | De-En | En-Fr | Fr-En | En-Ro | Ro-En | En-Ru | Ru-En | En-Zh | Zh-En | Avg |
| *Bilingual models* | | | | | | | | | | | |
| Transformer | 27.77 | 31.55 | 38.80 | 37.35 | 33.01 | 33.59 | 28.22 | 29.89 | 23.60 | 24.05 | **30.78** |
| GLAT | 26.09 | 30.53 | 38.62 | 34.44 | 31.83 | 32.59 | 25.42 | 28.13 | 21.05 | 22.89 | 29.20 |
| *Multilingual models* | | | | | | | | | | | |
| M-Transformer | 23.14 | 29.38 | 35.19 | 34.07 | 34.36 | 35.26 | 24.63 | 29.14 | 16.63 | 20.17 | 28.19 |
| CLSR | 21.57 | 30.01 | 34.07 | 32.72 | 30.61 | 35.29 | 19.13 | 30.27 | 16.28 | 20.19 | 27.01 |
| Adapter | 23.26 | 29.87 | 38.79 | 40.03 | 30.42 | 32.01 | 22.87 | 26.38 | 18.29 | 20.31 | 28.22 |
| MNAT | 8.33 | 13.86 | 12.07 | 19.21 | 12.89 | 21.36 | 7.66 | 15.43 | 4.85 | 6.01 | 12.16 |
| *switch*-GLAT | 24.18 | 30.49 | 39.47 | 36.30 | 31.93 | 32.40 | 24.16 | 28.33 | 16.25 | 21.23 | **28.47** |
| – w/o glancing | 6.83 | 16.29 | 11.08 | 20.33 | 13.57 | 23.34 | 7.57 | 18.25 | 7.31 | 6.74 | 13.13 |
| – w/o CSBT | 22.48 | 27.91 | 31.62 | 33.55 | 29.89 | 32.41 | 22.43 | 26.48 | 15.33 | 18.32 | 26.04 |
| – w/o CCS | 22.25 | 28.13 | 34.01 | 33.62 | 30.16 | 33.07 | 21.68 | 26.53 | 14.92 | 19.07 | 26.34 |

Table 2: Translation performance (BLEU) on WMT-many[2]. Avg means the average score.

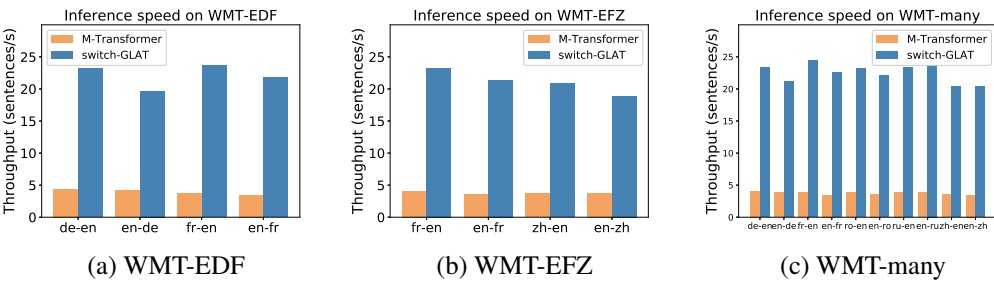

(a) WMT-EDF     (b) WMT-EFZ     (c) WMT-many

Figure 3: Inference speed evaluated on throughputs.

## 4.3 TRAINING AND INFERENCE DETAILS

We use 6 layers for encoder and parallel decoder. The model hidden size $d_{model}$ and feed-forward hidden size $d_{ff}$ are set to 512 and 2048 respectively. The number of attention head is set to 8. All the datasets are tokenized and segmented into subword units using BPE encodings (Sennrich et al., 2016). The vocabulary size is set to 85k for WMT-EDF and 95k for WMT-EFZ/many. The changing point E is set to $300,000$ steps and sampling number $S$ is set to $300,000$ for each pair. The mini-batch size is set to 64k tokens and the maximum training step is $1,200,000$. The model is trained with 8 NVIDIA Tesla V100 GPU cards. We follow the default parameters of Adam optimizer (Kingma & Ba, 2014) and learning rate schedule in Vaswani et al. (2017). Dropout annealing strategy (Rennie et al., 2015) is applied to stable training and the initialized dropout rate is set to $0.3$. In training, data from different language pairs are sampled according to a multinomial distribution rebalanced by a temperature of $0.3$ (Conneau et al., 2019).

Following the previous works (Gu et al., 2018; Li et al., 2020; Ma et al., 2019), the sequence-level knowledge distillation is used for all datasets, and the Transformer-base architecture (Vaswani et al., 2017) is employed as the teacher model. In inference, following Qian et al. (2020), *switch*-GLAT performs one-iteration parallel decoding. Moreover, we also leverage the common practice of noise parallel decoding (Gu et al., 2018) and the number of length reranking candidate is set to 5.

## 4.4 MAIN RESULTS

### 4.4.1 RESULTS ON WMT-EDF/EFZ

The left and right halves of Table 1 respectively show the results of WMT-EDF and WMT-EFZ.

---

[2]We updated all results in camera ready version using *multi-bleu.perl* script of Moses for evaluation because we misused *-lc* command in the original submission. Though the average BLEU scores were $0.0 \sim 0.5$ points lower than before, the overall conclusions are not changed.

***switch*-GLAT outperforms all multilingual competitors on the average score in both settings.** Specifically, in WMT-EDF scenario, our model achieves the best average BLEU score of 33.34 among all multilingual competitors, especially exceeding multilingual Transformer with a fairly significant margin of 0.35 point. Moreover, it performs better than CLSR and Adapter with an improvements of 1.33 and 1.26 points respectively. The same phenomena can be observed in WMT-EFZ scenario where the language pairs are more distant on linguistics. Our interpretation is that *switch*-GLAT has the ability to produce context-dependent code-switched translations with the code-switch decoder. The augmented code-switched translations can help model align the generated tokens and the original ones into the same vector space due to their similar context information, thus boosting machine translation performance (Mikolov et al., 2013).

***switch*-GLAT can largely speed up decoding.** In WMT-EDF scenario, the decoding speed tested on Fr-En pair is reported in Table 1, which is evaluated in the throughput of each model (full speed values are illustrated in Figure 3 (a) and Figure 3 (b)). Table 1 shows that the decoding speed of *switch*-GLAT is 6.2 times faster than multilingual transformer. Additionally, Figure 3 (a) and (b) also illustrate that the throughput of our model is significantly larger than multilingual transformer on all directions. These validate that the *switch*-GLAT can greatly improve decoding efficiency thanks to its parallel decoder. Note that the length predictor is jointly trained incorporating source and target language embeddings, thus leading to different length preference in different models.

***switch*-GLAT performs better than GLAT on average.** In both scenarios, *switch*-GLAT achieves higher performance than GLAT on average, especially X-to-En directions. These indicate that multilingual training makes it possible to transfer knowledge so that performance of X-to-En direction is enhanced due to the augmented data in English on the target side.

### 4.4.2 RESULTS ON WMT-MANY

Table 2 shows the results on the gathered WMT dataset from 10 language pairs.

***switch*-GLAT performs better than multilingual baselines in most cases.** It is primarily because *switch*-GLAT is able to produce diverse contextual code-switched translations, which encourage the model to automatically align the generated word vectors and the original ones due to their similar context information. Well-aligned representations are beneficial for improving machine translation performance. Notably, Adapter performs much worse than multilingual Transformer on En-Ro and En-Ru directions as the original paper interprets that Adapter do not perform well in low-resource settings (the reverse directions are enhanced by the English data from other language pairs).

***switch*-GLAT performs slightly worse than GLAT.** In this scenario, the failure of *switch*-GLAT may be owing to the limitation of model capacity. However, *switch*-GLAT achieves better or comparable performance with GLAT when the model translates from a foreign language to English. One interpretation is that all the parallel corpora including English on either source or target side can lead to a better parameter estimation of the English decoder (Firat et al., 2016).

***switch*-GLAT can enlarge the power of efficient decoding.** The decoding speed in this setting are illustrated in Figure 3 (c), showing that the throughput of our model is much larger than multilingual transformer on 10 directions. It demonstrates that *switch*-GLAT can enlarge the power of decoding efficiency since all translation directions can be accelerated within a single model.

### 4.5 ABLATION STUDY

The last three lines in Table 1 and Table 2 show the results of ablation study.

**Glancing sampling strategy plays an important role in *switch*-GLAT.** After removing the glancing sampling strategy, the performance drops most. It validates that glancing sampling is critical in training an advanced parallel multilingual translation model, which makes it suitable to generate high-quality code-switched translations.

**Code-switch back-translation promotes multilingual translation.** Excluding code-switch back-translation or employing code-switched data constructed by dictionary substitution both harms translation performance, of which the former hurts more. It validates that applying code-switched data can significantly increase multilingual translation performance, especially those generated by incorporating context information instead of dictionary replacing.

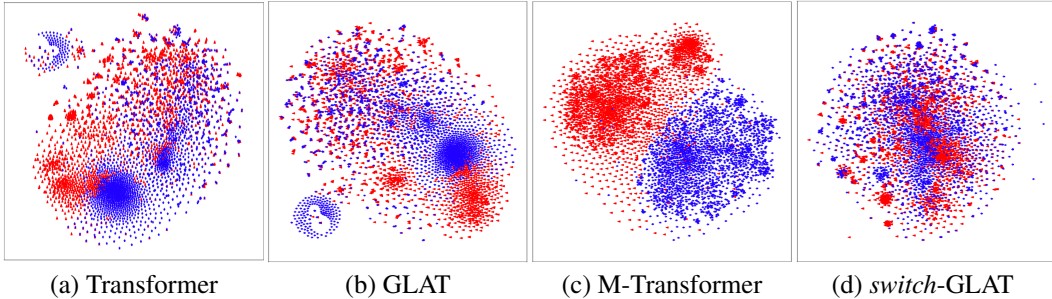

| (a) Transformer | (b) GLAT | (c) M-Transformer | (d) *switch*-GLAT |

Figure 4: Representations learned by (a) Transformer (Vaswani et al., 2017), (b) GLAT (Qian et al., 2020), (c) M-Transformer and (d) *switch*-GLAT, projected to 2D.

| Models | Word Induction | | | | | Sentence Retrieval | | | | |
|---|---|---|---|---|---|---|---|---|---|---|
| | En-De | De-En | En-Fr | Fr-En | Avg | En-De | De-En | En-Fr | Fr-En | Avg |
| M-Transformer | 30.2 | 31.7 | 32.7 | 38.9 | 33.3 | 23.6 | 22.3 | 24.5 | 23.8 | 23.5 |
| CLSR | 24.9 | 25.8 | 37.5 | 36.2 | 31.1 | 19.6 | 18.3 | 26.8 | 27.9 | 23.2 |
| Adapter | 32.8 | 34.5 | 40.1 | 42.8 | 37.5 | 34.6 | 28.7 | 26.9 | 35.2 | 31.4 |
| MNAT | 24.1 | 23.3 | 32.4 | 33.6 | 28.3 | 13.8 | 17.6 | 18.2 | 16.3 | 16.5 |
| *switch*-GLAT | 33.8 | 36.2 | 41.9 | 46.3 | **39.5** | 34.8 | 36.3 | 36.9 | 37.2 | **36.3** |
| – w/o glancing | 24.3 | 24.6 | 32.9 | 33.7 | 28.9 | 18.2 | 22.8 | 19.2 | 20.3 | 20.1 |
| – w/o CSBT | 28.2 | 30.1 | 31.9 | 37.8 | 32.0 | 17.6 | 21.4 | 23.3 | 25.3 | 21.8 |
| – w/o CCS | 30.5 | 29.7 | 33.6 | 35.8 | 32.4 | 18.8 | 22.9 | 22.7 | 21.6 | 21.5 |

Table 3: Results of quality analyses. Avg means the average accuracy.

## 5 ANALYSIS

In this section, we will make a rigorous analysis to reveal the reasons why our *switch*-GLAT can achieve better translation performance than multilingual transformer. All tasks are tested in WMT-EDF scenario.

### 5.1 VISUALIZATION

**There is much smaller gap in the cross-lingual representations learned by our *switch*-GLAT.** To intuitively illustrate how well the cross-lingual representations are aligned, we plot a two-dimensional projection of representations learned by four models in Figure 4. Here multilingual models are trained on the merged WMT-EDF dataset, while bilingual ones on the English → German part. We use the t-SNE algorithm (van der Maaten Laurens & Hinton, 2008) to perform the projection. English and German words are gold word pairs from Open Multilingual WordNet (OMW) datasets (Bond & Foster, 2013). English words are displayed in blue color and German words in red. We find there is much more overlapping between blue and red areas in Figure 4 (d), validating that *switch*-GLAT can produce better-aligned cross-lingual representations than other models.

### 5.2 CROSS-LINGUAL WORD INDUCTION

**The cross-lingual representations are aligned well at word level.** We assess the cross-lingual word induction performance to see how well the similar words from different languages are close to each other in the learned vector space (Gouws et al., 2015). Specifically, golden word pairs are extracted from the OMW datasets (Bond & Foster, 2013), resulting in $10,585$ En-De and $12,511$ En-Fr pairs. Top-1 accuracy is shown in the left half of Table 3. It shows that *switch*-GLAT significantly outperforms all autoregressive multilingual competitors on average, indicating that cross-lingual representations in our model are aligned well at word level.

### 5.3 PARALLEL SENTENCE RETRIEVAL

**The cross-lingual representations are aligned well at sentence level.** To further assess the quality of the cross-lingual representations at sentence level, we employ parallel sentence retrieval (PSR) task. PSR aims to extract parallel sentences from a comparable corpus between two languages. Following the settings of XTREME (Hu et al., 2020), we use the Tatoeba dataset (Artetxe & Schwenk, 2019). Cosine similarity is leveraged to search the nearest neighbour and the search accuracy of each model is reported in the right half of Table 3. It can be seen that *switch*-GLAT achieves the highest accuracy among all competitors, validating that better contextualized embeddings are obtained in *switch*-GLAT as the sentence embeddings are calculated with regarding all tokens in it.

## 6 RELATED WORK

### 6.1 MULTILINGUAL NEURAL MACHINE TRANSLATION

Early works extend the NMT model proposed by Bahdanau et al. (2014) to a multilingual version based either on LSTM (Hochreiter & Schmidhuber, 1997) or GRU (Cho et al., 2014). Dong et al. (2015) propose a one-to-many translation model (from English into 4 languages) by adding a dedicated decoder per target language, showing improvements over strong single-pair baseline. Zoph & Knight (2016) present a multi-source NMT which significantly outperforms a single-source model. Firat et al. (2016) propose a many-to-many model involving 10 language pairs by leveraging separate encoder and decoder per language as well as a sharing attention mechanism. Further, Ha et al. (2016) and Johnson et al. (2017) propose to train a multilingual model with a shared encoder-decoder-attention network across all languages to perform many-to-many translation. Recently, Transformer (Vaswani et al., 2017) is applied as the backbone of MNMT by many works. The Multi-Distillation (Tan et al., 2019) is trained to simultaneously fit the training data and match the outputs of individual bilingual models through knowledge distillation. Bapna et al. (2019) present to inject a language-specific adaptation layer into a pretrained model to keep the language-specific features. Aharoni et al. (2019) propose a multilingual NMT model involving up to 102 languages which achieves good zero-shot performance. All the above MNMT systems are based on autoregressive architectures which follow a sequential generation process and are limited to slow decoding speed.

### 6.2 NON-AUTOREGRESSIVE NEURAL MACHINE TRANSLATION

Recently, NAT has attracted much attention due to its ability of parallel decoding. A fully non-autoregressive model is first proposed by Gu et al. (2018), based on which latent variables are introduced to deal with dependencies among target words (Ma et al., 2019; Bao et al., 2019; Ran et al., 2021). Other branches of fully non-autoregressive systems either try to transfer knowledge from autoregressive models (Li et al., 2018; Wei et al., 2019; Guo et al., 2020; Sun & Yang, 2020), or apply different training objectives (Libovický & Helcl, 2018; Shao et al., 2020; Ghazvininejad et al., 2020; Qian et al., 2020). Besides, some works build NAT systems with structured prediction to model inter-dependencies among output words. Sun et al. (2019) propose to incorporate a structured inference module into the non-autoregressive models. Deng & Rush (2020) present a Markov transformer to perform cascaded decoding. Additionally, serveral semi-autoregressive models are also constructed to refine the outputs with multi-pass iterative decoding (Lee et al., 2018; Gu et al., 2019; Ghazvininejad et al., 2019; Kasai et al., 2020). All these works are limited to bilingual translation tasks. To our best knowledge, *switch*-GLAT is the first one to perform multilingual neural machine translation based on a non-autoregressive architecture.

## 7 CONCLUSION

This paper proposes *switch*-GLAT, a non-autoregressive multilingual neural machine translation model. It incorporates a switch-decoder, which can produce contextualized code-switched sentences and perform code-switch back-translation. Through this process, the multilingual translation performance and cross-lingual representations can both be improved. Besides, its parallel decoder enables a highly efficient inference. Experiments on three datasets suggest that *switch*-GLAT is superior to multilingual transformer in both effectiveness and efficiency. Further analyses also demonstrate the effectiveness of the learned contextual cross-lingual representations.

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

# A APPENDIX

## A.1 COMPARING WITH VANILLA BACK-TRANSLATION

We conduct ablation study of using vanilla back-translation (*switch*-GLAT-with-vanilla BT) in Table 4. It shows that though the model using vanilla back-translation outperforms the one not using any augmentation strategy (*switch*-GLAT-w/o-CSBT), it can be further improved by leveraging code-switch back-translation, validating that our proposed code-switch back-translation is favorable.

| Models | WMT-EFZ | | | | |
|---|---|---|---|---|---|
| | En-Fr | Fr-En | En-Zh | Zh-En | Avg |
| *switch*-GLAT | 40.54 | 36.48 | 19.47 | 22.55 | **29.76** |
| – with vanilla BT | 38.68 | 35.73 | 19.23 | 21.69 | 28.83 |
| – w/o CSBT | 35.17 | 33.65 | 18.32 | 20.37 | 26.87 |

Table 4: Ablation study of using vanilla back-translation on WMT-EFZ.

## A.2 APPLYING CSBT TO AUTOREGRESSIVE MODEL

We also apply our proposed code-switch back-translation (CSBT) to multilingual transformer, and the results are shown in Table 5. It shows that CSBT also boosts the performance of multilingual Transformer. Though M-Transformer-with-CSBT performs slightly better than our switch-GLAT, our model could significantly improve inference efficiency.

| Models | WMT-many | | | | | | | | | | |
|---|---|---|---|---|---|---|---|---|---|---|---|
| | En-De | De-En | En-Fr | Fr-En | En-Ro | Ro-En | En-Ru | Ru-En | En-Zh | Zh-En | Avg |
| M-Transformer | 23.14 | 29.38 | 35.19 | 34.07 | 34.36 | 35.26 | 24.63 | 29.14 | 16.63 | 20.17 | 28.19 |
| – with CSBT | 24.27 | 29.46 | 36.89 | 34.65 | 34.28 | 35.32 | 25.01 | 28.63 | 17.98 | 20.08 | **28.66** |
| *switch*-GLAT | 24.18 | 30.49 | 39.47 | 36.30 | 31.93 | 32.40 | 24.16 | 28.33 | 16.25 | 21.23 | 28.47 |
| – w/o CSBT | 22.48 | 27.91 | 31.62 | 33.55 | 29.89 | 32.41 | 22.43 | 26.48 | 15.33 | 18.32 | 26.04 |

Table 5: Translation performance of applying code-switch back-translation to multilingual Transformer.

## A.3 TRANSFER ABILITY

To validate that our model can improve the performance of low-resource languages, we train our *switch*-GLAT on the subset of TED dataset that is merged from English (En)-Spanish (Es), English-French (Fr) and English-Portuguese (Pt) pairs. En-Es and En-Fr pairs have about four times as much data as En-Pt pair. The results are shown in Table 6. It can be seen that our switch-GLAT can significantly improve the performance of low-resource languages (Pt-En) with as much as 19.91 and 0.95 BLEU points respectively compared with GLAT and Transformer.

To further validate the zero-shot ability of our model, we evaluated it (trained on WMT-many) on the test set of WMT 2019 German-French. Table 7 shows that the zero-shot ability of switch-GLAT and multilingual Transformer are both poor, but applying self pivot to these two models can achieve good performance.

| Models | TED | | | | | | |
|---|---|---|---|---|---|---|---|
| | En-Es | Es-En | En-Fr | Fr-En | En-Pt | Pt-En | Avg |
| Transformer | 34.62 | 38.52 | 37.78 | 36.34 | 27.79 | 30.94 | **34.33** |
| GLAT | 34.87 | 37.89 | 36.64 | 36.58 | 10.08 | 11.98 | 28.00 |
| *switch*-GLAT-w/o-CSBT | 34.03 | 38.27 | 35.62 | 36.79 | 21.18 | 30.54 | 32.74 |
| *switch*-GLAT | 34.86 | 38.69 | 36.04 | 37.92 | 23.58 | **31.89** | 33.83 |

Table 6: Translation performance of TED subsets.

| Models | | WMT-many | | |
|---|---|---|---|---|
| | | German-French | French-German | Avg |
| M-Transformer | zero-shot | 2.86 | 3.00 | 2.93 |
| | self pivot | 30.38 | 21.87 | **26.13** |
| *switch*-GLAT | zero-shot | 2.09 | 2.58 | 2.34 |
| | self pivot | 27.86 | 19.54 | 23.70 |

Table 7: Zero-shot performance of our *switch*-GLAT. Self pivot on German-French denotes first German to English and then English to French. So does French-German.

## A.4 RESULTS WITH A BIG MODEL

As capacity is crucial for multilingual models, we conduct experiments to show how *switch*-GLAT performance compares as model size grows. We adopt a standard Transformer-large architecture (Vaswani et al., 2017), which employs 6-layer encoder, 6-layer decoder and 1024 dimension on 16 heads. The results are shown in Table 8. It shows that in a multi-direction scenario, GLAT performs better than *switch*-GLAT when trained on a base model, but increasing model size brings opposite results, demonstrating that model capacity is quite critical for multilingual models.

## A.5 APPENDED VISUALIZATION

We also visualize the representations learned by CLSR since it is also a representative multilingual baseline. Again, we find there is much more overlapping between blue and red areas in Figure 5 (d), validating that *switch*-GLAT can produce better-aligned cross-lingual representations than other models.

## A.6 HYPER-PARAMETER SELECTION

We tune the training step of changing lambda (100000 to max steps with step size 100000), sample number S (100000 to max data size with step size 200000) and initialized dropout rate (0.1 to 0.5

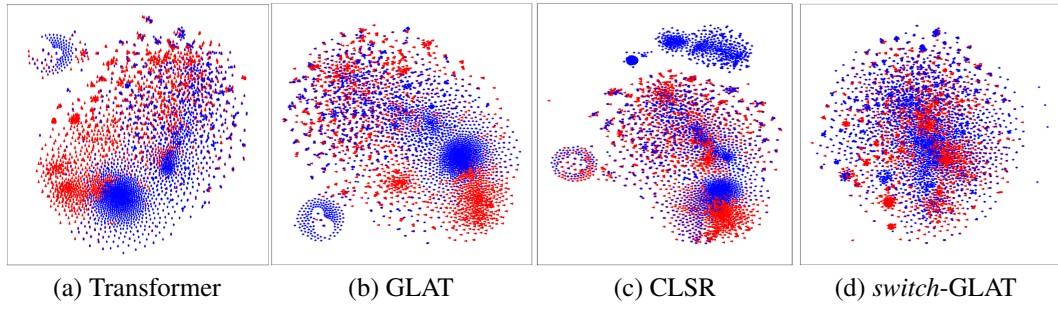

| (a) Transformer | (b) GLAT | (c) CLSR | (d) *switch*-GLAT |

Figure 5: Representations learned by (a) Transformer, (b) GLAT, (c) CLSR and (d) *switch*-GLAT, projected to 2D.

| Models | WMT-many | | | | | | | | | | |
|---|---|---|---|---|---|---|---|---|---|---|---|
| | En-De | De-En | En-Fr | Fr-En | En-Ro | Ro-En | En-Ru | Ru-En | En-Zh | Zh-En | Avg |
| *Base models* | | | | | | | | | | | |
| GLAT | 26.09 | 30.53 | 38.62 | 34.44 | 31.83 | 32.59 | 25.42 | 28.13 | 21.05 | 22.89 | 29.20 |
| *switch*-GLAT | 24.18 | 30.49 | 39.47 | 36.30 | 31.93 | 32.40 | 24.16 | 28.33 | 16.25 | 21.23 | 28.47 |
| *Large models* | | | | | | | | | | | |
| GLAT | 26.55 | 31.18 | 39.15 | 35.53 | 31.70 | 31.55 | 27.01 | 29.40 | 19.50 | 21.61 | 29.32 |
| *switch*-GLAT | 25.69 | 31.87 | 41.86 | 36.33 | 33.02 | 34.16 | 25.41 | 29.85 | 18.39 | 21.82 | **29.84** |

Table 8: Results of different model size on WMT-many.

with step size 0.1)with grid search on the validation set of WMT-EDF, and found that the values described in the paper performed best.

| $P_M$ | *switch*-GLAT | *switch*-GLAT-w/o-CCS |
|---|---|---|
| 0.0 | The longest journey begins with the first step. | The longest journey begins with the first step. |
| 0.1 | The longest voyage begins with the first step. | The longest journey begins with the first étape. |
| 0.3 | Le plus long viaje begins with the first step. | The longest journey empezar with the premier pisar. |
| 0.5 | Die längste Reise empieza con the first step. | The höchste viajar empezar with das first step. |

Table 9: Some examples of generated code-switched sentences respectively by our switch-GLAT and dictionary replacement. French, Spanish and German are represented in red, purple and yellow respectively.

## A.7 CASE STUDY

To gain an insight on how well our model can generate code-switched sentences, we provide some examples in Table 9. It shows that constructing code-switched sentences by dictionary replacement may cause errors. For example, *pisar* is a verb while the original word *step* is a noun, and using a verb here is not appropriate.

