# OpenReview forum: "switch-GLAT: Multilingual Parallel Machine Translation Via Code-Switch Decoder"
_ICLR.cc/2022/Conference — ICLR 2022 Poster_

### Official Review · Reviewer_ip8L · 2021-10-30

**Correctness:** 3
**Technical Novelty And Significance:** 3
**Empirical Novelty And Significance:** 3
**Recommendation:** 8
**Confidence:** 4

**Main Review:**

**Strengths**:
1. The paper presents a multilingual solution of NAT. The method works with GLAT and provides good insights on how to learn a good cross-lingual representation for NAT.


2. The proposed method speeds up the decoding process of the multilingual Transformer (M-Transformer). The generation quality of the proposed method is close to the bilingual autoregressive models (Transformer).


3. The author conducted comprehensive analytical studies on the improvement of the proposed method. It is clear to see how well the learned representations are aligned from the cross-lingual perspective. Besides, the word induction and parallel sentence retrieval task support the alignment ability in word-level and sentence-level, respectively.

**Weaknesses**:
1. The proposed method builds on the top of the GLAT method, which limits its generality. According to the ablation study, the performance dropped a lot without using the glancing mechanism (w/o glancing in Table 1 and 2).


2. Although I can understand that how the proposed method works, the description text of Figure 2 seems to be problematic. In Section 3.1, 3.2, the authors said that the code-switch decoder and code-switch back-translation process are illustrated in the left and middle modules, respectively. It is quite confusing that why they do not match the upper annotations shown in Figure 2.


3. There may be an error of mathematical symbol in Section 3.3. What does $L_{CS}$ refer to? Should it be $L_{bt}$?


4.    Missed one result of representation visualization. I understand that the adapter cannot learn cross-lingual alignment well because it models language-specific capability. Since CLSR can learn the shared pathway and language-specific capability together, why the authors did not choose to compare it in Section 5.1?


5.    The explanation that why switch-GLAT performs slightly worse than GLAT does not support by the experiments. The authors claim that it “may be owing to the limitation of model capacity”, will a bigger model or a balanced training set be helpful to alleviate this problem? It should be verified by additional experiments.


6. In Section 3.2, the authors said that “In the ﬁrst iteration of $P_M$, the number of mixed languages is set to 1. Afterward, it will be increased to one-third of the total.” There are four languages in WMT-EDF and WMT-EFZ training sets, does it mean that the number of mixed languages is always set to 1 (4*1/3~=1) during the training process? In this case, how the cross-lingual alignment is learned?


**Summary Of The Paper:**

The paper proposed a novel method for the multilingual non-autoregressive machine translation (NAT) model. The key idea is to modify the decoder module of glancing Transformer (GLAT), making it generate the contextual code-switched translations, then use them to perform code-switch back-translation. To incorporate with the generation of contextual code-switch translations, the authors proposed to add token-level language tags to the first layer and the final output layer at each position. The empirical results show that: 1) the inference speed is faster than the autoregressive baseline model (M-Transformer). 2) the translation performance outperforms the NAT baselines in terms of WMT-EDF, WMT-EFZ benchmarks. Although the performance is slightly worse than the GLAT model on the WMT-Many benchmark, the author claim that the reason is due to the limitation of model capacity. 3) Two cross-lingual experiments and the visualization of representation prove that the proposed method has better cross-lingual capability compared to other multilingual models.

**Summary Of The Review:**

The paper is well-written and contributes a novel approach for multilingual translation in NAT. The design of the code-switch back-translation is reasonable for boosting the performance of multilingual translation and the empirical analysis is convincing. Experiments show that switch-GLAT is better than other NAT/multilingual baselines in limited benchmarks and has close performance compared to bilingual AT baseline.

---

> ### Author Response · Authors · 2021-11-16
> **Response to Reviewer ip8L**
>
> Thanks to reviewer for helping improve our paper!  We have clarified all your  concerns, added the experiments and updated the paper accordingly.   We address the specific concerns as follows:
>
> **Q1: The proposed method builds on the top of the GLAT method, which limits its generality.**
>
> A1: We have applied code-switch back-translation to vanilla NAT (switch-GLAT-w/o-glancing). In Table1 and Table2, this model performs better than vanilla multilingual NAT with as much as 2.79 BLEU score, demonstrating that code-switch back-translation works on other NAT models as well. While code-switch back-translation is a general idea, it can improve the performance of switch-GLAT up to 3.20 BLEU points as the glancing strategy in training makes it adaptable to generate high-quality code-switch translations.
>
> **Q2: Since CLSR can learn the shared pathway and language-specific capability together, why the authors did not choose to compare it in Section 5.1?**
>
> A2: Thanks for the valuable comment. We have added the visualization of CLSR in Figure5, and again find there is much more overlapping between blue and red areas in Figure 5(d), validating that switch-GLAT can produce better-aligned cross-lingual representations than other models.
>
> **Q3: The explanations why switch-GLAT performs slightly worse than GLAT does not support by the experiments. Will a bigger model or a balanced training set be helpful to alleviate this problem? It should be verified by additional experiments.**
>
> A3: Thanks for the valuable suggestion. We have added the experimental results for big model in Table 9. It shows that increasing model size, switch-GLAT performs better than GLAT with 0.19 BLEU points on average, which is an opposite result to base models.  It demonstrates that model capacity is quite critical for multilingual models.
>
> | Methods | En-De | De-En | En-Fr | Fr-En | En-Ro | Ro-En | En-Ru | Ru-En | En-Zh | Zh-En | Avg |
> |  -  | -  | -  | - | -  | - | - | -  | -| -| - | -  |
> |GLAT (base) | 26.55 |31.02| 38.62| 34.44| 32.87| 33.51| 25.42| 28.13| 21.05| 22.89| 29.46|
> | switch-GLAT (base) | 24.22 | 31.04 | 39.11 | 36.10 | 32.18 | 32.85 | 24.03 | 28.21 | 16.70 | 21.52 | 28.60|
> |GLAT (big) |26.67|31.42|40.07|35.66|33.83|33.45|26.51|28.75|20.37|22.43|29.81|
> |switch-GLAT (big) |25.62|31.68|40.13|36.51|33.71|34.97|26.63|29.64|18.72|22.41|**30.00**|
>
> **Q4: There are four languages in WMT-EDF and WMT-EFZ training sets, does it mean that the number of mixed languages is always set to 1 (4*1/3~=1) during the training process? In this case, how the cross-lingual alignment is learned?**
>
> A4: There are only three languages in WMT-EDF and WMT-EFZ scenarios, and thus only one language could be used to generate code-switched sentences apart from source and target languages. Intuitively, training with augmented code-switched data can make model automatically align the generated words and the original ones into the same vector space according to their similar context information.
>
> We really hope our response addresses your concern. If you have any other questions, we are very happy to continue discussions!
>
> We notice that you have changed the score. If there are any new comments from you, we would be glad to hear them.
>
> We look forward for any further reading and feedback!

---

> > ### Author Response · Authors · 2021-11-21
> > **Looking forward to any other questions!**
> >
> > Thanks for your valuable reviews and comments again!
> >
> > If you have any other questions, please feel free to ask us!

---

> > > ### Comment · Reviewer_ip8L · 2021-11-27
> > > **Sounds convincing to me, thanks!**
> > >
> > > The authors have addressed my conerns and the new version sounds convincing to me. Thanks for your response.
> > >
> > > To follow-up, I still have some questions about the details, but this does not affect the contributions of this work:
> > >
> > > - $\lambda$ is used to control the amount of complex data to be introduced into the training process. How do you define the complexity level of the data (i.e., difficulty)? Have you analyzed what kinds of code-switched data are more useful, in terms of the number of languages and switches, particularly in the experiments of WMT-many?
> > >
> > >
> > > - In the WMT-many experiments, the _switch_-GLAT model was trained on datasets with 10 languages of high- and low-resource, can you elaborate the order of data used in training the model, i.e., high-resource languages first or random?

---

> > > > ### Author Response · Authors · 2021-11-27
> > > > **Thanks for your response!**
> > > >
> > > > Thanks for the reviewer's response! The answers to follow-up questions are as follows:
> > > >
> > > > **Q1: lambda is used to control the amount of complex data to be introduced into the training process. How do you define the complexity level of the data (i.e., difficulty)? Have you analyzed what kinds of code-switched data are more useful, in terms of the number of languages and switches, particularly in the experiments of WMT-many?**
> > > >
> > > > A1.1: We think that a code-switched sentence incorporating more languages and higher switching ratio is more complex since the decoder should gather meaning from more than one language and then translate the gathered semantics into the target language. This process requires the representations of the target language to align well with all the mixing languages.
> > > >
> > > > A1.2: Taking WMT-many as an example, we found that the code-switched data with three languages and cyclically changed switching ration were more useful for training a better model. Involving more than three languages or using a fixed switching ratio would hurt the translation performance.
> > > >
> > > > **Q2: In the WMT-many experiments, the switch-GLAT model was trained on datasets with 10 languages of high- and low-resource, can you elaborate the order of data used in training the model, i.e., high-resource languages first or random?**
> > > >
> > > > A2: At each training step, we randomly sample a language pair and then sample a batch from this pair to train switch-GLAT. In the first 150,000 steps, we sample a language pair from a uniform distribution to balance the performance of low-resource languages. After that, we sample a pair from a multinomial distribution as in [1] with q = p^(alpha) where p is the data size ratio and alpha = 0.3 .
> > > >
> > > > [1] Unsupervised Cross-lingual Representation Learning at Scale. Conneau et al. ACL 2020.

---

### Official Review · Reviewer_np9D · 2021-11-03

**Correctness:** 3
**Technical Novelty And Significance:** 2
**Empirical Novelty And Significance:** 3
**Recommendation:** 5
**Confidence:** 4

**Main Review:**

Strengths:

The authors examined the effectiveness of non-autoregressive NMT on multilingual translation, and obtained encouraging supervised translation performance on several data conditions.


Weaknesses:

* Some claims and proposals are not well supported by the current experiments;
* Experimental details are not always clear to me;
* Model comparison might be unfair and misleading, and more comparisons are required.


Comments:

1)	Firstly, at the core of the adaptation of the GLAT model to multilingual translation is the newly proposed code-switch decoder. This decoder uses token-level target language tag to produce code-switch translations. But the necessity of such design is not tested. Why use code-switch decoder not the vanilla GLAT decoder? A comparison to a vanilla multilingual GLAT should be given.
2)	Also, the authors propose code-switch back-translation to improve model’s performance, but don’t prove why such specific BT is favorable or necessary. What if using the vanilla BT to the NMT model? For example, this can be achieved via random online back-translation proposed in [1].
3)	The authors show that the proposed model improves cross-lingual representation compared to auto-regressive NMT models, but results in Table 3 show that BT delivers a large impact on such improvement without which switch-GLAT performs much worse. Then a question would be: what if applying BT to auto-regressive baselines as well (such as the above mentioned random online BT)?
4)	 Some experimental details are very unclear! How did you train your baseline models, CLSR and Adapter? What’s the hyperparameters? For CLSR, what’s your budget constraint? How did you set it? In addition, how did you train the MNAT baseline? Did you apply the recently introduced optimization techniques as in [2]?
5)	You mentioned in page 5 that knowledge distillation is used. What’s your teacher model, bilingual Transformer base or multilingual Transformer base? Did you also apply the KD to all baselines (autoregressive and non-autoregressive)? I would assume that the authors used bilingual models for KD and only applied it to the NAT models. In such a case, claiming multilingual NAT “outperforms” multilingual AT is non-convincing.
6)	As explained in the introduction, multilingual NMT benefits low-resource translation, which also enables zero-shot translation. Neither are explored in the current experiments. WMT also offers low-resource languages. Would switch-GLAT benefit these languages in a multilingual setup? What about switch-GLAT’s zero-shot ability?

[1] Zhang et al., 2020 Improving Massively Multilingual Neural Machine Translation and Zero-Shot Translation
[2] Gu et al., 2020 Fully Non-autoregressive Neural Machine Translation: Tricks of the Trade


**Summary Of The Paper:**

This paper extends non-autoregressive NMT models, particularly the GLAT model, from bilingual translation to multilingual translation. To enable such extension, the authors proposed token-level language tags for the decoder, code-switch decoder paired with back-translation, and also scheduled training between the traditional MLE loss and the BT loss. Experiments on several multilingual tasks based on WMT corpus show that the proposed model obtains better supervised translation quality than its auto-regressive counterparts and delivers better cross-lingual representations.

**Summary Of The Review:**

In short, this paper explores an interesting direction: applying non-autoregressive models to multilingual translation. But the experiments suffer from unclarity, unfair comparison and over claims. The paper could be largely improved with more analysis and comparisons.

---

> ### Author Response · Authors · 2021-11-16
> **Response to Reviewer np9D**
>
> We thank the reviewer for the positive and detailed review as well as the suggestions for improvement. We have clarified all your  concerns, added the experiments and updated the paper accordingly. Our response to the reviewer’s question is below:
>
> **Q1: Why use code-switch decoder not the vanilla GLAT decoder? A comparison to a vanilla multilingual GLAT should be given.**
>
> A1: Thanks for your comment. The comparison to a vanilla multilingual GLAT has been given in Table1 and Table2, which we call switch-GLAT-w/o-CSBT. From the two tables, we can see that our proposed switch-GLAT performs better than the vanilla multilingual GLAT with as much as 3.06 BLEU score.
>
> **Q2: What if using the vanilla BT to the NMT model?**
>
> A2: Thanks for this suggestion!  switch-GLAT using code-switch back-translation (CSBT) performs better than using vanilla BT with 1.24 BLEU score on WMT-EFZ dataset, validating the effectiveness of our proposed CSBT.
>
> | Methods | En-Fr | Fr-En | En-zh | Zh-En | Avg |
> |  -  | -  | -  | - | -  | - |
> |switch-GLAT | 41.79|37.33|19.35|23.69|**30.54**|
> |switch_GLAT with vanilla BT  | 39.39 | 36.63 |19.21 | 22.37|29.30|
> |switch_GLAT w/o CSBT | 35.78|34.22|18.37|21.01|27.34|
>
> **Q3: what if applying BT to auto-regressive baselines as well ?**
>
> A3: Thanks for this valuable suggestion! We have added the experiments for applying code-switch back-translation (CSBT) to auto-regressive baseline (Multilingual Transformer). The results show that multilingual Transformer can be improved with 0.55 BLEU score by using CSBT, while our switch-GLAT can be improved with 2.06 BLEU, which is much larger than autoregressive model. Though M-Transformer-CSBT performs slightly better than our switch-GLAT, our model could significantly improve inference efficiency.
>
> | Methods | En-De | De-En | En-Fr | Fr-En | En-Ro | Ro-En | En-Ru | Ru-En | En-Zh | Zh-En | Avg |
> |  -  | -  | -  | - | -  | - | - | -  | -| -| - | -  |
> |M-Transformer | 23.22 | 29.67 | 35.51 | 34.11 | 34.40 | 35.82 | 24.68 | 29.25 | 16.59 | 20.61 | 28.39|
> |M-Transformer-CSBT|29.96|24.52|37.34|34.93|34.65|35.72|25.02|29.17|17.82|20.30|**28.94**|
> |switch_GLAT w/o CSBT | 22.89 | 28.78 | 32.22 | 34.31 | 30.31 | 33.06 | 22.46 | 27.13 | 15.27 | 9.06 | 26.54|
> |switch-GLAT | 24.22 | 31.04 | 39.11 | 36.10 | 32.18 | 32.85 | 24.03 | 28.21 | 16.70 | 21.52 | **28.60** |
>
> **Q4: Some experimental details are very unclear!**
>
> A4: We will make it more clear. We reimplemented Adpater following the released code in [1]. We applied the released code of CLSR and extended it to a many2many version by adding a target language tag at the beginning of target sentence. We built MNAT baseline by adding token-level language tags to both encoder and decoder of vanilla NAT, and trained it like M-Transformer.
>
> [1] Counter-Interference Adapter for Multilingual Machine Translation. Zhu.et al. EMNLP 2021.
>
> **Q5: What’s your teacher model, bilingual Transformer base or multilingual Transformer base? Did you also apply the KD to all baselines (autoregressive and non-autoregressive)?**
>
> A5: We use the bilingual Transformer as the teacher model and apply KD to all the NAT models. We believe NAT will achieve comparable results to AT models without using KD in the near future.
>
> **Q6: Would switch-GLAT benefit low-resource languages in a multilingual setup? What about switch-GLAT’s zero-shot ability?**
>
> A6: Thanks for the valuable suggestion! We have added experiments on low-resource and zero-shot transfer. The results show that our switch-GLAT can significantly improve the performance of low-resource languages (Pt-En) with as much as 20.14 and 1.18 BLEU points respectively compared with GLAT and Transformer.  Besides, zero-shot abilities (trained on WMT-many, tested on WMT19 German-French) of switch-GLAT (2.34) and multilingual Transformer (2.94) are both poor, but applying self pivot to these two models can achieve good performance (switch-GLAT 23.94, M-Transformer 26.48).
>
> **Experiments on low-resource performance.**  We test switch-GLAT on the Ted subsets merged from En-Es, En-Fr and English-Pt pairs, of which the number of En-Es and En-fr are four times that of En-Pt. The results show that our switch-GLAT can significantly improve the performance of Pt-En respectively compared with GLAT (+20.14) and Transformer (+1.18).
>
>  |Methods| En-Es | Es-En | En-Fr |Fr-En | En-Pt | Pt-En |Avg|
> |-|-|-|-|-|-|-|-|
> |Transformer |34.62|38.52|37.78|36.34|27.79|30.94|**34.33**|
> |GLAT |34.87|37.89|36.64|36.58|10.08|11.98|28.00|
> |switch-GLAT-w/o-CSBT|34.61|38.55|35.83|37.28|21.61|31.55|33.23|
> |switch-GLAT|35.02|39.01|36.58|38.12|23.78|**32.12**|34.10|
>
> We really hope our response addresses your concern. If you have any other questions, we are very happy to continue discussions!
>
> We look forward for any further reading and feedback!
>
> If you have any other questions, please free free to ask us!

---

> > ### Author Response · Authors · 2021-11-21
> > **Looking forward to any other questions**
> >
> > Thanks for your valuable review and comments again.
> >
> > If you have any other questions, please feel free to ask us

---

> > > ### Comment · Reviewer_np9D · 2021-11-25
> > > **Thanks for your response! Some follow-up questions**
> > >
> > > Thanks for your response! I have some follow-up questions!
> > >
> > > 1) Did you use the sentence-level language tag or token-level tag for `switch-GLAT-w/o-CSBT`?
> > > 2) How did you apply the vanilla BT to switch-GLAT? Did you also use and tune the schedule for $\lambda$ in Eq. (9)?
> > > 3) Did you apply the token-level tag to the multilingual Baseline? How did you perform CSBT for M-Transformer?
> > > 4) If you apply KD from bilingual models to NAT, using it to AT models could make the comparison fair. As in Table 1, bilingual models still largely outperform multilingual models on En->X translation.
> > > 5) Could you please show more details about the low-resource multilingual setup? training/valid/test statistics, and also which datasets are used for evaluation? Also results for the used bilingual teacher model.
> > > 6) Regarding zero-shot translation, the zero-shot quality of all models in Table 7 is very poor, which tells little. This might be because the authors used sentence-level tags on both the source and target sides for M-Transformer, which could greatly hurt the zero-shot performance, see [1]. Or, perhaps WMT-many has too diverse languages that hinder zero-shot transfer. Could you show the De-Fr result for WMT-EDF as well?
> > >
> > > [1] Language Tags Matter for Zero-Shot Neural Machine Translation

---

> > > > ### Author Response · Authors · 2021-11-26
> > > > **Thanks for your response!**
> > > >
> > > > Thanks for reviewer's response and valuable suggestion! Answers to follow-up questions are as follows:
> > > >
> > > > **Q1: Did you use the sentence-level language tag or token-level tag for switch-GLAT-w/o-CSBT?**
> > > >
> > > > A1: Yes, to make the comparison fair, we used token-level language tag for switch-GLAT-w/o-CSBT. Therefore, the only difference between switch-GLAT-w/o-CSBT and our switch-GLAT is whether to use the code-switch back-translation.
> > > >
> > > > **Q2: How did you apply the vanilla BT to switch-GLAT? Did you also use and tune the schedule for in Eq. (9)?**
> > > >
> > > > A2: To apply vanilla BT to switch-GLAT, we followed the traditional back-translation, taking English-German pair as an example: first we generated synthetic German targets using the En2De AT model, and then reversed the pair to get pseudo parallel data <generated German, English source>.  Actually, it's like a reverse KD process. As [1] validated that adding raw data prior can improve translation performance, applying vanilla BT to switch-GLAT is superior to switch-GLAT-w/o-CSBT. We also used the scheduling strategy for switch-GLAT with vanilla BT, which means that vanilla BT was applied to switch-GLAT after E steps (300,000 in our paper) to make the comparison fair.
> > > >
> > > > [1] Understanding and Improving Lexical Choice in Non-Autoregressive Translation. Ding et al. ICLR 2021.
> > > >
> > > > **Q3: Did you apply the token-level tag to the multilingual Baseline? How did you perform CSBT for M-Transformer?**
> > > >
> > > > A3: Yes, to make comparison fair, we applied token-level language tag to all multilingual baselines except CLSR and Adapter. The released code in CLSR only used an indicator at the beginning of the source sentence, and we extended it to a many2many version by adding another indicator at the beginning of target sentence. We reimplemented Adpater following the released code in [2] which used a sentence-level language tag. Given the token-level language tags at each position in advance, M-Transformer can also generate code-switched sentences and do CSBT. Specifically, we first sample a target length from the uniform distribution U(L-5, L+5) where L is the source sentence length, and then do CSBT routine as switch-GLAT except that the generation process is autoregressive.
> > > >
> > > > [2] Counter-Interference Adapter for Multilingual Machine Translation. Zhu.et al. EMNLP 2021.
> > > >
> > > > **Q4: If you apply KD from bilingual models to NAT, using it to AT models could make the comparison fair. As in Table 1, bilingual models still largely outperform multilingual models on En->X translation.**
> > > >
> > > > A4: Thanks for your suggestion. The born-again networks in [3] proved that applying KD to AT models did not yield better performance, but we will add this setting to make the comparison fair. For bilingual models outperform multilingual models on En-X directions, the possible reason may be language discrepancy and interference as well as limited model capacity.
> > > >
> > > > [3] Understanding knowledge distillation in non-autoregressive machine translation. Zhou et al. ICLR 2020.
> > > >
> > > > **Q5: Could you please show more details about the low-resource multilingual setup? training/valid/test statistics, and also which datasets are used for evaluation? Also results for the used bilingual teacher model.**
> > > >
> > > > A5.1: We used the English (En) - Spanish (Es), English - French (Fr)  and English - Portuguese (Pt) pairs (including training/valid/test splits) from TED'13 datasets, which are released by [4]. The detailed statistics are as follows:
> > > >
> > > > |Pairs|Training|Valid |Test|
> > > > |-|-|-|-|
> > > > |En-Es |196026|4231|5571|
> > > > |En-Fr |192304|4320|4866|
> > > > |En-Pt|51785|1193|1803|
> > > >
> > > > [4] When and Why Are Pre-Trained Word Embeddings Useful for Neural Machine Translation? Qi et al. NAACL 2018.
> > > >
> > > > A5.2: The results in low-resource setting are shown in the following table and the Transformer is used as the AT teacher. The table shows that switch-GLAT can significantly improve the performance of low-resource pair (Pt->En) with as much as 20.14 and 1.18 BLEU points respectively compared with GLAT and Transformer, validating that multilingual training has the ability to improve the low-resource translation performance.
> > > >
> > > >  |Methods| En-Es | Es-En | En-Fr |Fr-En | En-Pt | Pt-En |Avg|
> > > > |-|-|-|-|-|-|-|-|
> > > > |Transformer |34.62|38.52|37.78|36.34|27.79|30.94|**34.33**|
> > > > |GLAT |34.87|37.89|36.64|36.58|10.08|11.98|28.00|
> > > > |switch-GLAT|35.02|39.01|36.58|38.12|23.78|32.12|34.10|
> > > >
> > > > **Q6: The zero-shot quality of all models is very poor, which tells little. Could you show the De-Fr result for WMT-EDF as well?**
> > > >
> > > > A6: The following table shows the zero-shot results of De-Fr on WMT-EDF scenario. As we can see, the zero-shot ability in this setting shows similar phenomena to WMT-many, which is poor. It might be the language tags on both source and target sides hurt the transfer ability as you said.
> > > >
> > > > |Methods| De-Fr | Fr-De |Avg|
> > > > |-|-|-|-|
> > > > |M-Transformer zero-shot |2.15|2.30|2.23|
> > > > |M-Transformer self pivot |27.08|23.94|**25.87**|
> > > > |switch-GLAT zero-shot|2.05|2.06|2.05|
> > > > |switch-GLAT self pivot|23.57|22.34|22.94|

---

> > > > > ### Comment · Reviewer_np9D · 2021-11-27
> > > > > **Something to be clarified**
> > > > >
> > > > > Thanks for your response!
> > > > >
> > > > > 1) Based on my experience, a normal multilingual transformer could perform very well on zero-shot De-Fr on WMT-EDF. I believe that the zero-shot result reported here is misleading and adds little. But since the switch operation or token-level language tag is required by switch-GLAT, one could conclude that switch-GLAT doesn't deliver zero-shot transfer.
> > > > >
> > > > > 2) How did you train your multilingual baselines?
> > > > >    - You mentioned explicitly in the paper that **We set a multilingual transformer baseline, which uses a sentence-level language tag on both source and target sides.**, but in the response, you claim  that **we applied token-level language tag to all multilingual baselines except CLSR and Adapter**.
> > > > >    - In my understanding, the back-translation used by CSBT is done by switch-GLAT, right? but why use En2De AT model for the vanilla BT to switch-GLAT for comparison? I think you could do the vanilla BT by disabling the *switch operation*, i.e. always using the same target language token, but why not? Using an extra AT model for BT might introduce capacity mismatch that results in better or worse performance, and makes the comparison unfair.
> > > > >
> > > > > 3) The study [1] mainly focuses on bilingual translation settings. I don't think it has direct evidence to support that applying KD from bilingual AT models to multilingual AT models helps little.
> > > > >
> > > > > [1] Understanding knowledge distillation in non-autoregressive machine translation. Zhou et al. ICLR 2020.
> > > > >
> > > > > 4) About low-resource translation, what's the "GLAT" baseline? In the paper, GLAT is mainly explained as a bilingual baseline, but here I assume it's a multilingual baseline. So, how did you adapt GLAT to be multilingual? what's the language tag? how did you apply BT? Is language tag the only difference between GLAT and switch-GLAT? did you also apply KD to GLAT?

---

> > > > > > ### Author Response · Authors · 2021-11-27
> > > > > > **Thanks for pointing out the problem and we clarify the confusing description**
> > > > > >
> > > > > > **Comments 1:  The confusing description about language tag for multilingual Transformer**
> > > > > >
> > > > > > A1: We are really sorry for the confusing description. At first we trained a multilingual Transformer with sentence-level language tags on both source and target sides. Later, to make the comparison fair, we also tried token-level language tag for M-Transformer and found that it performed better than sentence-level one as shown in the following table. So the M-Transformer results reported in our paper are achieved by using token-level one. Afterwards, we would proofread the whole paper to guarantee that this problem will not occur. We are really sorry for the misleading description and thanks for helping point out it.
> > > > > >
> > > > > > |Methods|En-De|De-En|En-Fr|Fr-En|Avg|
> > > > > > |-|-|-|-|-|-|
> > > > > > |M-Transformer with sentence-level tags|24.63|30.00|39.02|35.67|32.36|
> > > > > > |M-Transformer with token-level tags|26.21|31.61|38.45|35.77|33.01|
> > > > > >
> > > > > > **Comments 2: Based on my experience, a normal multilingual transformer could perform very well on zero-shot De-Fr on WMT-EDF. I believe that the zero-shot result reported here is misleading and adds little.**
> > > > > >
> > > > > > A2: Thanks for reviewer's valuable comments. As study [1] shows, using language tags on both source and target sides sometimes yields extremely bad zero-shot results according to specific settings and thus I guess that the token-level tag on both source and target sides hurt the zero-shot performance. Multilingual training could improve the performance of low-resource languages or provide zero-shot transfer ability. We verified that our proposed switch-GLAT can improve the low-resource performance on the Pt->En pair from the merged TED'13 subset. Since zero-shot transfer is not our main goal, how to improve the zero-shot performance in our switching and token-level tag setting will be surveyed in the future.
> > > > > >
> > > > > > [1] Language Tags Matter for Zero-Shot Neural Machine Translation. Wu et al. ACL 2021.
> > > > > >
> > > > > > **Comments 3: Why use En2De AT model for the vanilla BT to switch-GLAT for comparison? I think you could do the vanilla BT by disabling the switch operation, but why not? Using an extra AT model for BT might introduce capacity mismatch that results in better or worse performance, and makes the comparison unfair.**
> > > > > >
> > > > > > A3: Yes, you are right as a fairer comparison should be done when doing the vanilla BT by disabling the switch operation. Since generating back-translation data is time-consuming and we have the data generated by bilingual AT, so we did vanilla back-translation using this data due to the limited time. As switch-GLAT-w/o-CSBT performs worse than bilingual AT, I guess the back-translation data generated by bilingual AT would have higher quality than generated by switch-GLAT-w/o-CSBT, and using this data for vanilla BT performs worse than using CSBT, demonstrating that our proposed CSBT is superior than vanilla BT. We will do the experiments as you said to make comparison fairer.
> > > > > >
> > > > > > **Comment 4: The study [2] mainly focuses on bilingual translation settings. I don't think it has direct evidence to support that applying KD from bilingual AT models to multilingual AT models helps little.**
> > > > > >
> > > > > > A4: Yes, you are right and thanks for your advice. The born-again network in [2] focuses on bilingual models and it can't demonstrate that KD from bilingual AT model wouldn't benefit multilingual AT models. Later I will applying KD from bilingual AT model to multilingual AT baselines to make the comparison fair.
> > > > > >
> > > > > > [2] Understanding knowledge distillation in non-autoregressive machine translation. Zhou et al. ICLR 2020.
> > > > > >
> > > > > > **Q5: About low-resource translation, what's the "GLAT" baseline? In the paper, GLAT is mainly explained as a bilingual baseline, but here I assume it's a multilingual baseline. So, how did you adapt GLAT to be multilingual? what's the language tag? how did you apply BT? Is language tag the only difference between GLAT and switch-GLAT? did you also apply KD to GLAT?**
> > > > > >
> > > > > > A5: In the low-resource setting, the "GLAT" baseline is a bilingual baseline, so it don't incorporate any language tags. We also apply KD to GLAT and it performs poorly on Pt-En and En-Pt as GLAT don't perform well on small datasets. As the following table shows, Transformer denotes the bilingual Transformer which is used as the teacher model for KD, GLAT is a bilingual NAT model and switch-GLAT-w/o-CSBT is the multilingual GLAT not using CSBT. switch-GLAT-w/o-CSBT performs better than Transformer and GLAT on Pt-En respectively with 0.61 and 19.57 points, and switch-GLAT can further improve the performance, demonstrating that multilingual training could improve the low-resource performance.
> > > > > >
> > > > > > |Methods| En-Es | Es-En | En-Fr |Fr-En | En-Pt | Pt-En |Avg|
> > > > > > |-|-|-|-|-|-|-|-|
> > > > > > |Transformer |34.62|38.52|37.78|36.34|27.79|30.94|**34.33**|
> > > > > > |GLAT |34.87|37.89|36.64|36.58|10.08|11.98|28.00|
> > > > > > |switch-GLAT-w/o-CSBT|34.61|38.55|35.83|37.28|21.61|31.55|33.23|
> > > > > > |switch-GLAT|35.02|39.01|36.58|38.12|23.78|32.12|34.10|

---

> > > > > > ### Author Response · Authors · 2021-11-27
> > > > > > **Thanks for your detailed and valuable suggestions!**
> > > > > >
> > > > > > We are really sorry for the confusing description again. We think you are a responsible and professional reviewer and give us many valuable comments and suggestions, which helps improve our paper a lot. We will carefully proofread the whole paper to guarantee that this problem will not occur.

---

### Official Review · Reviewer_cdKo · 2021-11-03

**Correctness:** 3
**Technical Novelty And Significance:** 3
**Empirical Novelty And Significance:** 3
**Recommendation:** 6
**Confidence:** 4

**Main Review:**

Strengths:
- Novel application of NAT to a multilingual setting, with an interesting new strategy for code-switched back-translation.
- Impressive gains over a fairly broad selection of baselines, including strong auto-regressive models.
- Potential to be a significant breakthrough in increasing both speed and accuracy of multilingual models.

Weaknesses:
- The capacity used is quite small for multilingual models. It certainly seems sub-optimal for the WMT-many setting, where both bilingual baselines are starting to pull away from the multilingual models. Since capacity is such a crucial parameter for multilingual models, experiments to show how this new technique compares as models grow would have made the paper more convincing.
- There are two new things in this paper: the application of NAT to a multilingual setting, and the use of code-switched back-translation. It would have been interesting to tease them apart using NAT architectures other than GLAT: how well do other recent NAT models work on multilingual data; how much do they benefit from code-switched back-translation (since that idea should apply to any NAT model)? The NAT baseline used in the paper is not competitive.
- There are many hyper-parameters involved in the setup. No information is supplied about how carefully they were tuned, or how they would generalize to new settings.
- Some problems with clarity, see details below.

Details:
- This is hard to understand: “	Such efficiency problem is more serious in multilingual setting because serving many translation directions in one model always leads to a higher throughput practically.”
- This statement should be qualified: “NAT generates translation outputs in parallel (Gu et al., 2018), which leads to significantly faster translation speed.” See: Kasai, Jungo, et al. "Deep encoder, shallow decoder: Reevaluating non-autoregressive machine translation." arXiv preprint arXiv:2006.10369 (2020).
- s3.2 isn’t very clear. The code-switched generation (middle panel in figure 2) seems to have abandoned the GLAT training idea, and just be the standard CMLM algorithm with language tags added to the masks to get code-switched output?
- s4.2 How exactly does switch-GLAT-w/o-glancing work?
- 4.4.2 "switch-GLAT is superior to all multilingual baselines on the average BLEU score.” - not really, it’s basically tied with Adapter and M-Transformer.


**Summary Of The Paper:**

This paper describes experiments in training a non-autoregressive Transformer (glancing Transformer, or GLAT) in a multilingual MT setting. The use of a non-autoregressive model, plus token-level language tags, enables a code-switching strategy where a source sentence in a given language can generate a target sentence containing words from multiple specified languages. This is used to augment the multilingual training corpus with back-translated data consisting of code-switched synthetic source sentences paired with original monolingual target sentences. A multilingual GLAT model trained with this strategy is shown to outperform many baselines including monolingual GLAT and several autoregressive multilingual models. Ablation studies demonstrate that code-switched back-translation is essential for achieving good performance.


**Summary Of The Review:**

The first application of non-autoregressive transformers to the important problem of multilingual MT, with a novel idea (code-switched back-translation) specific to that application. Results are generally impressive, but the experiments are missing crucial model-scaling data points, and they miss the opportunity to evaluate code-switched back-translation independent from the GLAT architecture the authors favour.

---

> ### Author Response · Authors · 2021-11-16
> **Response to Reviewer cdKo**
>
> We thank the reviewer for the positive review and insightful questions.  We have clarified all your  concerns, added the experiments and updated the paper accordingly.  Answers to specific points are below:
>
> **Q1: Since capacity is such a crucial parameter for multilingual models, experiments to show how this new technique compares as models grow would have made the paper more convincing.**
>
> A1: Thanks for the valuable suggestion. We have added the experimental results for big model in Table 9. It shows that increasing model size, switch-GLAT performs better than GLAT with 0.19 BLEU points on average, which is an opposite result to base models.  It demonstrates that model capacity is quite critical for multilingual models.
>
> | Methods | En-De | De-En | En-Fr | Fr-En | En-Ro | Ro-En | En-Ru | Ru-En | En-Zh | Zh-En | Avg |
> |  -  | -  | -  | - | -  | - | - | -  | -| -| - | -  |
> |GLAT (base) | 26.55 |31.02| 38.62| 34.44| 32.87| 33.51| 25.42| 28.13| 21.05| 22.89| 29.46|
> | switch-GLAT (base) | 24.22 | 31.04 | 39.11 | 36.10 | 32.18 | 32.85 | 24.03 | 28.21 | 16.70 | 21.52 | 28.60|
> |GLAT (big) |26.67|31.42|40.07|35.66|33.83|33.45|26.51|28.75|20.37|22.43|29.81|
> |switch-GLAT (big) |25.62|31.68|40.13|36.51|33.71|34.97|26.63|29.64|18.72|22.41|**30.00**|
>
> **Q2 : how well do other recent NAT models work on multilingual data; how much do they benefit from code-switched back-translation ?**
>
> A2: We have applied code-switch back-translation to vanilla NAT (switch-GLAT-w/o-glancing). In Table1 and Table2, this model performs better than vanilla multilingual NAT with as much as 2.79 BLEU score, demonstrating that code-switch back-translation works on other NAT models as well. While code-switch back-translation is a general idea, it can improve the performance of switch-GLAT up to 3.20 BLEU points as the glancing strategy in training makes it adaptable to generate high-quality code-switch translations.
>
> **Q3: No information is supplied about how carefully hyper-parameters were tuned**
>
> A3: We tune the training step of changing lambda (100000 to max steps with step size 100000), sample number S (100000 to max data size with step size 200000) and initialized dropout rate (0.1 to 0.5 with step size 0.1)with grid search on the validation set of WMT-EDF, and found that the values stated in the paper performed best.
>
> **Q4: The code-switched generation seems to have abandoned the GLAT training idea, and just be the standard CMLM algorithm with language tags added to the masks to get code-switched output?**
>
> A4: Yes, the generation process proceeds as you described and it can be regarded as CMLM in generation. In training and code-switch back-translation process, switch-GLAT still follow the GLAT training criteria.
>
> **Q5: How exactly does switch-GLAT-w/o-glancing work?**
>
> A5: switch-GLAT-w/o-glancing denotes the model that applies code-switch back-translation to multilingual vanilla NAT. In Table1 and Table2, this model performs better than vanilla multilingual NAT with as much as 2.79 BLEU score, demonstrating that code-switch back-translation works on other NAT models as well.
>
> We really hope our response addresses your concern. If you have any other questions, we are very happy to continue discussions!
>
> We look forward for any further reading and feedback!

---

> > ### Comment · Reviewer_cdKo · 2021-11-19
> > **thanks**
> >
> > Thanks for the response.
> >
> > For Q1, these results are quite convincing, but for an eventual paper you would want to show increased-capacity results for all models, not just these two.
> >
> > For Q2, I realize that you compared to a version of vanilla NAT already. I was interested in knowing how the code-switching technique works with other recent NAT models, since GLAT has many competitors. The idea was to make the paper more about multilingual NAT + code-switching in general, less about multilingual GLAT. Or, you could show that there's something special about GLAT that makes it uniquely benefit from code-switched BT, but that might be harder.

---

> > > ### Author Response · Authors · 2021-11-21
> > > **Thanks for your response**
> > >
> > > Thanks for your valuable comments!
> > >
> > >     1. **To comments1**:
> > > Thanks for the valuable comments. From the above results, we can that switch-GLAT outperforms GLAT after improving the model capacity, which can validate the claim that switch-GLAT  performs worse than GLAT in WMT-many is due to the limited capacity. But as you suggest, we would give increased-capacity results for all models.
> > >
> > >     2. **To comments2**:
> > > Thanks for the insightful comments. Applying code-switch batck-translation to a vanilla multilingual NAT can boost its performance, which demonstrates that our proposed code-switch back-translation is not limited to GLAT. The glancing strategy makes the training and inference process consistent as they both produce remained words conditioned on the glanced words, which makes GLAT more appropriate to do code-switch back-translation. As you can see, swich-GLAT can improve with as much as 3.20 BLEU points compared with not using it.

---

### Comment · Area_Chair_aFzB · 2021-11-14
**Additional Discussion Encouraged**

Dear Reviewers,

can you please take a look at each other's reviews? Your reviews currently straddle the decision boundary and it would be good to make sure you have considered all the perspectives provided. Please update your reviews (at least to acknowledge that you have read all reviews).

Thanks,
Your Area Chair

---

> ### Comment · Reviewer_np9D · 2021-11-29
> **Discussion After Author Rebuttal**
>
> Hi all,
>
> After checking the authors' responses, I'm still not convinced by the big claim "*non-autoregressive models are better multilingual translators*", although I agree that the focused research direction, combining multilingual modeling and non-autoregressive modeling, is interesting and deserves more future efforts.
>
> Multilingual NMT often gives the following benefits:
> 1) support translation for multiple languages and produce decent (even improved) quality for supervised language pairs;
> 2) help low-resource languages greatly via knowledge transfer from high-resource languages
> 3) enable zero-shot transfer and deliver zero-shot translation for unseen training language pairs
>
> Based on current experimental results and further information from the authors' rebuttal, I have the following concerns:
> 1) non-autoregressive multilingual model still underperforms its autoregressive counterpart, even the former was trained with knowledge distillation from bilingual or multilingual autoregressive teachers (Table 5).
> 2) the knowledge transfer to low-resource languages is still worse than autoregressive models, again note that switch-GLAT also used knowledge distillation (see Table 6, Transformer vs. switch-GLAT w/o CSBT).
> 3) the proposed switch-GLAT delivers very poor zero-shot translation results. Although the authors also show inferior results for multilingual autoregressive baselines in Table 7, this is caused by abnormal language tagging (the authors acknowledge this), and the comparison is unfair (using a clearly weak baseline for comparison tells little).
>
> Besides, some important details are missing or confusing in the experiments and some comparisons are highly likely unfair though. Either way, I'm still not convinced that a non-autoregressive multilingual translator is better. I keep my original judgment!
>
> best regards,
> Reviewer np9D

---

> > ### Author Response · Authors · 2021-11-29
> > **Responses to np9D**
> >
> > Thanks for reviewer np9D's comments! Our responses to the reviewer's concerns are listed as follows:
> >
> > 1. **For our title:**  The "better" means a trade-off between translation quality and inference speed. We use "better" to show that our model can achieve comparable or even better results and simultaneously much faster inference speed compared with multilingual Transformer. It seems that "better" is a little misleading here, and later we could change the title to make our target clearer.
> >
> > 2. **Compared with multilingual Transformer:** As shown in Table1 and Table2, our switch-GLAT can outperform multilingual Transformer with as much as 1.16 BLEU improvement and simultaneously 6.6x faster decoding speed in inference. In Table5, multilingual Transformer can also be improved by using our proposed code-switch back-translation. After using our proposed CSBT, though M-Transformer performs slightly better than our switch-GLAT with 0.34 BLEU score on average, our model can achieve about 6 times faster decoding speed.
> >
> > 3. **Low-resource language performance:** As shown in Table7, through multilingual training, switch-GLAT-w/o-CSBT can outperform GLAT and Transformer respectively with 19.57 and 0.61 BLEU points on low-resource direction Pt-En. switch-GLAT-w/o-CSBT can also perform significantly better than GLAT with 11.53 BLEU points on En-Pt. By using our proposed code-switch back-translation, the low-resource performance can be further improved. These demonstrate that our switch-GLAT can improve low-resource performance through positive knowledge transfer.
> >
> > 4. **Zero-shot performance:**  To apply our proposed code-switch back-translation, the token-level language tag is required. As study [1] shows, using language tags on both source and target sides sometimes yields extremely bad zero-shot results according to specific settings and thus I guess that the token-level tag on both source and target sides hurt the zero-shot performance.   In this paper, our main goal is to propose a multilingual NAT model with code-switch back-translation which can achieve comparable or even better performance and much faster decoding speed, as well as give some insight for further multilingual NAT research. How to improve the zero-shot performance in our code switching and token-level tag setting will be studied in the future.
> >
> > [1] Language Tags Matter for Zero-Shot Neural Machine Translation. Wu et al. ACL 2021.

---

### Author Response · Authors · 2021-11-16
**Summary of Revisions**

We thank all the reviewers' valuable suggestions.

We’ve uploaded a revised draft incorporating reviewer feedback. Below is a summary of the main changes:

* The corrected claim in Introduction (R1)
*The clarified writing of code-switch decoder and code-switch back-translation (R3)
*The experiments of switch-GLAT using vanilla back-translation in Table4(in appendix) (R2)
*The experiments of applying code-switch back-translation to autoregressive models in Table5 (in appendix) (R2)
*The experiments of low-resource transfer and zero-shot performance of switch-GLAT in Table6 and Table7 (in appendix) (R2)
*The experiments of big models in Table8 (in appendix) (R1 & R3)
*The visualization of CLSR in Figure 5 (R3)
*Adding a case study in Table9 (in appendix)

And we look forward to any further reading and feedback.

---

### Decision · Program_Chairs · 2022-01-20

**Decision:**

Accept (Poster)

**Comment:**

This paper proposes several innovations for machine translation. The reviewers had several questions about the claims that were made and the authors addressed these and also acknowledged that some of their formulations (e.g. 'better') would need to be qualified. Overall, there are several interesting ideas that have been put together in a sensible way, but the story is not super consistent.

The detailed exchanges between the reviewers are authors are commendable!